# Path Planning in Narrow Road Scenarios Based on Four-Layer Network Cost Structure Map

**DOI:** 10.3390/s25092786

**Published:** 2025-04-28

**Authors:** Ping Wang, Hao Zhang, Youming Tang

**Affiliations:** 1School of Mechanical and Automotive Engineering, Xiamen University of Technology, Xiamen 361024, China; pipiwang3487@xmut.edu.cn (P.W.); 18854851211@163.com (H.Z.); 2Key Laboratory of Advanced Design and Manufacturing of Passenger Vehicles of Fujian Province, Xiamen 361024, China; 3School of Mechanical and Energy Engineering, Zhejiang University of Science and Technology, Hangzhou 310023, China

**Keywords:** narrow roads, Voronoi layer, four-layer network structure, path planning, B-spline smoothing

## Abstract

To address the issues of insufficient safety distance and unsmooth paths in AGV path planning for narrow road scenarios, this paper proposes a method that integrates Voronoi-skeleton-based custom layers with traditional cost maps. First, key nodes of the Voronoi skeleton are extracted to generate a custom layer, which is then combined with static, obstacle, and expansion layers to form a new four-layer network cost map. This approach accurately distinguishes obstacle influences and enhances algorithm robustness. The A* algorithm based on this new map guides the automated guided vehicle (AGV) to travel safely along the road center. Second, an improved A* algorithm is employed for global planning to ensure safe navigation. Finally, B-spline smoothing is applied to the global path to enhance the AGV’s efficiency and stability in complex environments. The experimental results show that in narrow road scenarios, the proposed algorithm improves AGV path planning safety by 82%, reduces the number of spatial turning points by 55.85%, and shortens planning time by 48.98%. Overall, this algorithm significantly enhances the robustness and real-time performance of path planning in narrow roads, ensuring the AGV moves safely in an optimal manner.

## 1. Introduction

Automated guided vehicles (AGVs) are widely used in industry, agriculture, warehousing, logistics, search and rescue, and other fields. AGV path planning is a key technology for modern intelligent transportation and autonomous driving [1]. In warehousing and logistics, AGVs often operate in narrow aisle scenarios where various static obstacles pose a threat to safety, placing higher demands on navigation safety and mobility efficiency in such environments [2]. Traditional AGV global path planning algorithms include the A* algorithm [3], the Dijkstra algorithm [4], and the ant colony algorithm [5], while local path planning algorithms usually use the timed elastic band method [6] and the pure tracking algorithm [7].

In path planning research, narrow road scenarios usually refer to environments where the road width is small compared to the size of the AGV. To quantify this concept, this study defines “narrow road” as a ratio of road width to AGV width less than or equal to 2.0. That is, $R=\frac{W_{road}}{W_{AGV}}\leq 2.0$, where $W_{road}$ is the road width, and $W_{AGV}$ is the width of the AGV. Path planning in narrow road scenarios is mostly implemented using traditional algorithms. However, since the planned paths are prone to being close to obstacles and are not smooth, the efficiency of AGVs in actual operation is low and there are certain safety hazards.

For example, Tang Lina et al. [8] used radio frequency identification (RFID) technology to determine the position and location of AGV in a narrow road environment and proposed a minimum time path planning algorithm at any angle and any target position. Geng Xijun et al. [9] improved the device operating node (DQN) algorithm to improve the success rate of the path and reduce the path cost in a narrow road scenario. The above studies have achieved global path optimization in a narrow road environment but have not fully considered the problem of path safety distance. Especially in a complex environment, the safety distance between AGV and obstacles is not effectively guaranteed, and the path smoothness is poor, making it difficult to ensure the stable operation of AGV.

The cost map constructed using the Voronoi diagram can visually reflect obstacle distributions in the environment, providing effective support for path planning. The A* algorithm, with its advantages in search efficiency and path quality, can generate optimal global paths by incorporating known obstacle information [10]. Zohaib et al. [11] used the bubble bug algorithm (BBA), which is an enhancement of the intelligent bug algorithm (IBA) and improved obstacle avoidance and path efficiency through the bubble band technique (BBT) and bubble rebound algorithm (BRA). Hassan M U et al. [12] proposed a low-cost, autonomous robot with an innovative seed selector mechanism that improves efficiency while ensuring a low miss rate. This robot design emphasizes the optimization of hardware modules and trajectory tracking performance, demonstrating its potential for application in agricultural environments. The trajectory performance optimization in the above paper inspired this paper to propose that the safety of robot path planning in narrow constrained environments is a key factor. Xing Wu et al. [13] used the Voronoi skeleton for global path optimization and introduced a dynamic window approach (DWA) for guidance, but the DWA algorithm resulted in frequent turns in narrow passages, lowering AGV navigation efficiency. Zohaib M et al. [14] proposed the intelligent bug algorithm (IBA) as a goal-oriented improved algorithm. Compared with the traditional bug algorithm, it can achieve a shorter and smoother trajectory and complete the goal in a shorter time. The IBA algorithm effectively improves the convergence by introducing a bidirectional mechanism, combining sensor configuration and robot field of view (FOV), especially in trajectory smoothness and time efficiency. It is better than the Dist-Bug algorithm. However, the IBA algorithm has limitations when dealing with extremely narrow paths. Its core idea is to plan as close to obstacles as possible to avoid collisions. However, in extremely narrow road conditions, being too close to one side of the obstacle may cause a collision when turning, and it is impossible to ensure a sufficient safety distance. This problem limits its application effect in narrow environments. Hui Gong et al. [15] utilized geometric topology combined with Dijkstra’s algorithm for smoothing, optimizing the geometric shape of paths to maintain high computational efficiency in complex environments and improve AGV motion stability. Yilmaz S. et al. [16] proposed an improved bat algorithm (BA) that enhances its exploration and exploitation mechanisms through three modifications. The improved algorithm can obtain better solution quality when solving benchmark problems compared to the standard version. Almeida S. L. et al. [17] explored the construction and optimization of Voronoi structures, improving genetic algorithm performance through Voronoi structure optimization, analyzing trends in 2D and 3D Voronoi structure optimization, and proposing future research directions integrating additive manufacturing constraints. While these algorithms have their own advantages in path planning, they have not specifically addressed path planning in narrow road scenarios, leaving unresolved issues regarding AGV movement safety distance and path smoothness.

In summary, this paper proposes a Voronoi-skeleton-based AGV path planning method for AGV motion in complex environments such as narrow roads. First, key nodes of the Voronoi skeleton are extracted to generate a custom Voronoi layer, which is integrated with a three-layer cost map to iteratively form a new four-layer network cost map. Compared with the three-layer cost map, this approach provides a more accurate representation of obstacle influence, improving algorithm robustness and adaptability. Second, the A* algorithm based on the new cost map guides the AGV to travel along the road center, avoiding collisions with nearby obstacles and enhancing path planning safety. Third, B-spline optimization is applied to smooth the path and eliminate discontinuities, improving AGV stability and comfort in complex environments. Finally, a comparative path planning experiment is conducted in a real-world complex narrow road scenario to verify the proposed method’s safety, computational efficiency, and real-time performance.

## 2. Materials and Methods

### 2.1. Definition for Narrow Road Scenarios

In this study, “narrow road” is defined as a scenario where the road width is smaller than the size of the AGV. To quantify this criterion, the ratio of the road width $W_{road}$ and $W_{AGV}$, the AGV width is set as follows:(1)$$\begin{array}{c} R=\frac{W_{road}}{W_{AGV}}\leq 2.0 \end{array}$$

Among them, R represents the ratio of road width to *AGV* width. If *R* is less than or equal to a certain threshold $R_{threshold}$, then the road is considered a “narrow road”. In this study, the threshold selected is 2.0, When the ratio of road width to *AGV* width is less than or equal to 2.0, the environment will be considered a “narrow road” scenario, The schematic diagram is shown in Figure 1.

By defining and analyzing the “narrow road” scenario, this section clarifies the applicability and optimization methods of this study’s algorithm in such environments. Compared with traditional algorithms, this method pays more attention to the safe distance and path smoothness in narrow roads, ensuring that AGVs can operate stably and safely in these complex environments.

### 2.2. Voronoi Skeleton Description

The Voronoi diagram is a spatial partitioning method [18] that divides a space into multiple regions using a set of seed points. Each region is called a cell [19]. Within each cell, all points are closer to the seed point of that cell than to any other seed point. As shown in Figure 2, ρ(1) and ρ(2) represent dimensional space points. The boundaries of the Voronoi cells are the intersections of the perpendicular bisectors of the seed points. Using the Euclidean distance, the shortest distance from two points to the Voronoi triangular cell is calculated as X and Y. Due to these characteristics, the Voronoi diagram is well-suited for obstacle avoidance strategies in path planning, enhancing both the safety and robustness of the path [20].

For a given set of seed points $\left\{S_{1},S_{2},\ldots ,S_{n}|S_{i}\in R^{d},i=1,2,\ldots ,n\right\},$ any point in the space is assigned to the Voronoi cell corresponding to the seed point $S_{i}$, where $R^{d}$ represents the coordinate space of all seed points in a d-dimensional space. This assignment holds if and only if the condition in Equation (1) is satisfied:(2)$$\begin{array}{c} V_{i}=\left\{P\in R^{d}|\forall j\neq i,d\left(P,S_{i}\right)<d\left(P,S_{j}\right)\right\} \end{array}$$

Here, $d\left(P,S_{i}\right)<d\left(P,S_{j}\right)$ means that the distance from point P to any other seed point $S_{j}$ is greater than its distance to $S_{i}$ Only points satisfying this condition are included in the region associated with seed point $S_{i}$ Each Voronoi cell contains all points that are closest to its corresponding seed point, and its boundary is formed by points that are equidistant from the neighboring seed points. This characteristic ensures that each Voronoi path stays as far as possible from obstacles, allowing the AGV to navigate safely within the mapped environment.

In this paper, the calculation core of the Voronoi diagram is first based on the grid cost map of the surrounding environment as input. The cost map is generated by obtaining the point cloud data of the surrounding environment through sensors such as the LiDAR on the AGV and combining it with the mapping algorithm. Common mapping algorithms include Cartography and Gmapping. In this process, the environmental data obtained by the Laser scan are converted into a cost map, in which the cost value of each grid cell reflects the distribution of obstacles in the area. Obstacles usually correspond to areas with higher costs, while open areas correspond to lower cost values.

In the process of generating the Voronoi diagram, the selection of seed points is crucial. The seed point defines the “center” of each area in the Voronoi diagram and determines the shape of each area in the diagram. In this study, we selected the positions of static obstacles in the cost map as seed points. The locations of these obstacles usually represent the core areas that need to be avoided in the environment. By using these locations as seed points, it can be ensured that the generated Voronoi diagram can accurately reflect the impact of obstacles on path planning. In this way, the Voronoi diagram can provide effective environmental feedback to the AGV, ensure that obstacles are avoided during path planning, and effectively optimize the driving route.

### 2.3. New Cost Map of Four-Layer Network Structure

A traditional cost map is typically composed of a three-layer network structure, which generates a complete cost map by overlaying a static layer, an obstacle layer, and an inflation layer [21]. When applied to complex environments such as narrow road scenarios, this three-layer cost map structure often results in planned paths that are too close to obstacles due to factors such as high obstacle density, large map coverage, and a low proportion of feasible paths. This compromises navigation safety. Furthermore, when new obstacles appear in the environment, the inflation layer must be recalculated, leading to poor real-time performance.

To address these issues, as shown in Figure 3, this paper extracts key points from the Voronoi skeleton, generates a custom Voronoi layer, and merges it with the static layer, obstacle layer, and expansion layer to iteratively form a new cost map with a four-layer network structure. The new cost map accurately divides obstacles into a series of Voronoi regions, effectively characterizing the influence range of obstacles in complex environments such as narrow roads and avoiding overlap between obstacles and planned paths. In addition, the distance relationship between adjacent obstacles can be calculated in real time, thereby dynamically adjusting the path to achieve effective obstacle avoidance. Especially in complex narrow road environments, this method maximizes path safety.

In this study, the generation of the Voronoi diagram and the skeleton extraction process are used to improve path planning and provide effective obstacle avoidance capabilities for AGV. As shown in Figure 4, the whole process is mainly implemented through the following steps:(1)Generation of Voronoi Diagram: First, the Voronoi algorithm is used to generate the Voronoi diagram of the entire environment. In this process, we regard the static obstacles in the environment as seed points and divide these obstacles into multiple Voronoi regions through the Voronoi diagram. Each Voronoi region represents the shortest distance from any point in the region to the nearest obstacle, and the boundary of the region is the dividing line between the obstacle and the free space.(2)Image Binarization: After generating the Voronoi diagram, we convert each region in the diagram into a binary image. In this binary image, the obstacle location and the Voronoi skeleton are marked as “1”, while other regions are marked as “0”. This step simplifies the Voronoi diagram into a binary form, which is convenient for subsequent skeleton extraction and refinement.(3)Voronoi skeleton refinement: The thinning algorithm is applied to the binary image to extract the core structure of the Voronoi skeleton. The thinning process iteratively removes unimportant boundary points and retains only the most important paths connected to the core of the region. At each iteration, the algorithm removes isolated points or redundant line segments in the image, and the final retained path is the Voronoi skeleton. The skeleton consists of several line segments connecting obstacles and free space, showing the influence range of obstacles in the environment.(4)Extract key points: In the refined Voronoi skeleton, we extract key points, which represent the main structure of the skeleton and the bifurcation points of the path. The extracted key points are usually located at the intersection of the boundary between obstacles and free space, or at the location where the path direction changes greatly in the skeleton. These key points will be used as constraints in path planning to ensure that the AGV path avoids obstacles and provides a safe distance.(5)Generate custom Voronoi layers: Through the extracted key points, we generated a custom Voronoi layer. This layer is used to represent the impact area and safety distance of obstacles from the perspective of AGV path planning. The custom Voronoi layer is combined with the traditional cost map (including static layer, obstacle layer and expansion layer) to form a new four-layer network structure.(6)Integration in ROS system: The generated Voronoi diagram data are integrated into the system through the plugin mechanism in the ROS system. Specifically, the plugin is responsible for receiving the generated Voronoi diagram data and converting it into a layer that can be used for AGV navigation. The custom Voronoi layer is combined with the traditional cost map to form a four-layer network structure, including static layer, obstacle layer, inflation layer, Voronoi layer. In this way, the cost map of the four-layer network structure can more accurately reflect the impact range of obstacles and improve the safety and efficiency of path planning.

As shown in Figure 5, the new cost map interface is integrated into the ROS operating system, and the interaction steps are as follows: the new cost map is inherited under the layer manager and used as the output of the ROS system to manage the cost map; the extracted Voronoi skeleton key points independently generate custom layers and merge with the traditional cost map to iteratively form a new four-layer network structure, which relies on the ROS operating system to adjust the input and output; the extension layer of the new cost map is superimposed on the obstacle layer, and the center distance is determined to the greatest extent through the Voronoi skeleton characteristics, ensuring that obstacles can be avoided and a safe distance can be maintained to the greatest extent during navigation path planning.

### 2.4. Global Path Planning Based on New Cost Map

The traditional A* algorithm for path planning typically considers only the path length or optimal distance without taking path safety into account. The Voronoi algorithm proposed in this paper uses the new cost map as a constraint for the A* algorithm and performs a depth-first search to generate a globally optimized path with the safest distance. The specific process is as follows: To address the issues of insufficient safety distance and lack of path smoothness in the traditional A* algorithm, this paper proposes an improved A* algorithm with the cost function $f\left(n\right)=g\left(n\right)+h\left(n\right)$, where g(n) represents the actual cost from the start point to the current node, and h(n) is the estimated cost from the current node to the target. After integrating the Voronoi layer, the cost function can be further expanded as follows:(3)$$\begin{array}{c} f\left(n\right)=g\left(n\right)+h\left(n\right)+C\left(v\left(n\right)\right) \end{array}$$

In the equation, C(v(n)) represents the cost value of node *n* in the Voronoi layer, indicating the “quality” of the region. By observing the value of C(v(n)), the safety and reliability of the regional path can be determined. By incorporating the Voronoi layer into the heuristic function h(n), obstacle information is optimized, allowing the heuristic function to adjust based on obstacle distribution and cost information. This enhances the efficiency of the search process. During the global path search process, especially in complex narrow road environments, the proposed algorithm selects path nodes with lower costs for expansion. It follows the guidance of the Voronoi layer to avoid traversing high-cost areas. By utilizing the newly proposed cost map to reasonably partition safe regions, the global path can effectively avoid obstacles when computing the cost function.

Figure 6 shows the path planning results of the A* and improved A* algorithms on the same map. To generate this comparison chart, we used a randomly generated grid map for the experiment and controlled the experimental environment by setting simulation parameters. The specific experimental parameters and settings are as follows:(1)Setting the grid map size: The grid map used in this experiment is 20 × 20, that is, a two-dimensional matrix with 20 rows and 20 columns, where each cell represents a grid.(2)Obstacle setting: In path planning algorithms, obstacles are usually generated by random number generators. To ensure that a consistent obstacle layout is generated in each experiment, we use a random seed to fix the starting state of the random number generator. This experimental map is simulated in Python3.9, and the random seed in the map is set by calling the built-in seed function. In this way, the generated random number sequence is the same each time it is run, ensuring the repeatability of the obstacle layout. In this study, obstacles are randomly distributed on the map with a probability of 40%. Specifically, each grid cell has a 40% probability of becoming an obstacle, and the rest is empty space. To achieve this, we generate a random number between 0 and 1 for each grid cell. If the random number is less than or equal to the set obstacle probability, the grid cell will be marked as an obstacle; otherwise, it will be a feasible path. In this way, we can simulate an environment with random obstacles for path planning algorithms to test and optimize.(3)Start and end point settings: The starting point is set at the upper left corner of the grid (coordinate (1, 1)); The end point is set near the lower right corner of the grid (coordinates (18, 18)), that is, a certain distance from the lower right corner.

In the traditional A* algorithm, path planning relies only on the spatial information of the grid, ignoring the actual size and motion characteristics of the AGV. This may result in some planned paths being too narrow for the AGV to pass through. By introducing turning costs, we add additional constraints to path planning to ensure that the path not only avoids obstacles but also takes into account the AGV’s own volume, thereby avoiding planning a global path that requires “crossing gaps”. In the improved A* algorithm, the turning cost $C_{turn}(n)$ is mainly determined by calculating the angle change in the path. We introduce a cost for each turning point in the path, which is proportional to the angle change of the path, to ensure that the AGV can travel smoothly and stably during path planning. The specific turning cost calculation formula is as follows:(4)$$\begin{array}{c} C_{turn}\left(n\right)=\alpha \cdot angle\left(n\right) \end{array}$$

In the above formula, α is a constant coefficient that is used to adjust the weight of the turning cost and control the impact of the turning on the path cost; $angle\left(n\right)$ is the angle between node n and the nodes before and after it in the path, indicating the degree of turning of the path. This angle reflects the turning action required by the AGV when turning and is closely related to the motion characteristics and body size of the AGV.

By introducing the turning cost, we can not only take into account the additional cost of the AGV when turning but also ensure that the planned path will not pass through gaps between narrow obstacles or difficult-to-navigate areas, thereby maximizing the safety and stability of the AGV’s movement.

In summary, it can be seen from Figure 6 that the path generated by A* has more direction changes and obvious sharp turns; the path generated by the improved A* is smoother, avoids unnecessary sharp turns, and is more in line with the vehicle’s motion characteristics. A* only considers the shortest path and ignores the vehicle’s minimum turning radius. In autonomous driving and robot navigation tasks, the vehicle’s motion is constrained by physical properties and cannot make large directional changes. The improved A* solves this problem by introducing turning costs, making the path more in line with the vehicle’s motion characteristics and improving driving stability. On the other hand, the traditional A* algorithm is prone to falling into local optimality, making the path too dependent on the choice of the heuristic function h(n), resulting in a shorter but not smooth path selected during the search process. In contrast, the improved A* algorithm can find a more reasonable path by introducing a more complex $C_{turn}\left(n\right)$.

### 2.5. B-Spline Smoothing Optimization of Global Path

During AGV navigation, a global path is first generated based on the starting point and goal point using a priority search. However, the path generated by the new cost map with the four-layer network structure exhibits issues such as zigzagging and large turning angles. To address these issues, this paper uses an improved B-spline curve to smooth and optimize the global planning path based on the new cost map.

B-spline curves adjust their shape locally through control points and offer more flexible order settings compared to traditional Bézier curves. They allow independent specification of the curve’s degree and order, effectively avoiding excessive peaks and valleys and enabling local modifications to overcome the limitations of Bézier curves [22]. Specifically, when using B-spline curves, after specifying the degree, moving a control point only changes part of the shape without affecting the overall curve.

Let there be control points ${\;P}_{0},P_{1},P_{2}$, …, $P_{n+1}$, a total of n + 1 control points, which are used to define the direction and boundary of the spline curve. The definition of a k-th order B-spline curve with n + 1 control points is as follows:(5)$$\begin{array}{c} p\left(u\right)\left[P_{0},P_{1},\ldots ,P_{n}\right]=\left[\begin{array}{c} B_{0,k}\left(u\right)\\ B_{1,k}\left(u\right)\\ \vdots \\ B_{n,k}\left(u\right) \end{array}\right]\sum_{i=0}^{n}P_{i}B_{i,k}\left(u\right) \end{array}$$

In the formula, $B_{i,k}(u)$ is the i-th k-th order B-spline basis function, corresponding to the control point $P_{i}$, where k ≥ 1; u is the independent variable. The basis function follows the De Boor–Cox recursion formula:(6)$$B_{i,k}(u)=\{\begin{array}{cc} 1, & u_{i}\leq u<u_{i+1}\\ 0, & other \end{array}\left(k=1\right)$$(7)$$\begin{array}{c} B_{i,k}\left(u\right)=\frac{u-u_{i}}{u_{i+k-1}-u_{i}}B_{i,k-1}\left(u\right)+\frac{u_{i+k}-u}{u_{i+k}-u_{i+1}}B_{i+1,k-1}\left(u\right)\left(k\geq 2\right) \end{array}$$

If the denominator is 0, as shown in Figure 7, the following conventions are applied:

If the numerator is also 0, the entire term is defined as 0.

If the numerator is not 0, the denominator is conventionally set to 1.

In the equation, $u_{i}$ is a set of continuously varying values known as the knot vector, which is a non-decreasing sequence. The first and last values are typically defined as 0 and 1. The sequence is as follows:(8)$${[u}_{0},u_{1},u_{2},\ldots ,u_{k},u_{k+1},\ldots ,u_{n},u_{n+1},\ldots ,u_{n+k}]$$

As shown in Figure 8, the image illustrates the recursive computation process of B-spline basis functions and their influence across different knot intervals. The main content is as follows:

First, the domain is divided into multiple knot intervals, including $\left[u_{0},u_{1}\right]$, $\left[u_{1},u_{2}\right]$,…, $\left[u_{4},u_{5}\right]$. These intervals define the support range of the B-spline basis functions, with each interval affecting different basis functions.

Second, the image employs a triangular structure to represent the construction process of B-spline basis functions. Starting from the left side, each layer of basis functions $B_{i,k}\left(u\right)$ is computed recursively from the previous layer using the recurrence relation given in Equation (5) [23]. This equation demonstrates how higher-order B-spline basis functions are obtained through a linear combination of lower-order basis functions.

Finally, the points $P_{0}$ and $P_{1}$ on the right side indicate the influence range of the control points. As the order k increases, the basis functions become smoother and influence a larger number of control points.

As shown in Figure 9, this paper compares and analyzes the open, clamped, and closed types of uniform B-splines and quasi-uniform B-splines. As shown in Figure 9a, if the node vector has no specific structure, the generated curve does not connect at the beginning and end. This type of B-spline curve is called an open B-spline [24]. As shown in Figure 9b, by adjusting the parameter u within the domain $\left[u_{i},\;u_{i+1}\right]$, a clamped B-spline curve with fixed end points is generated. As shown in Figure 9c, by adjusting the number and position of the data nodes and control points, a closed B-spline curve is generated, where the starting and ending points of the curve form a closed loop. Therefore, based on the stability characteristics of the B-splines in the figure (Stability characteristics at the path planning level are mainly reflected in smoothness, continuity, and controllability.), the closed B-spline curve has the best smoothing performance. The reason is that closed B-splines can smoothly connect the path at the end points, making the transition between the start and end points of the path more natural, thus avoiding sharp turning points and unnecessary turns in the path.

However, for non-closed paths, although the start and end points of the path do not coincide, we can still use closed interval B-spline curves for smoothing. In this case, the advantage of closed B-spline is that it can ensure the overall smoothness of the path by introducing virtual connection points (the connection between the virtual start and end points) during the smoothing process. Specifically, by introducing additional control points at both ends of the path, the B-spline curve appears to be “closed”, thereby achieving a smooth path transition. These virtual points do not affect the actual start and end points of the path, but through the extension of the curve, the continuity and smoothness of the path are ensured.

Therefore, even if the path itself is open, we can still use the properties of closed B-spline to ensure the effect of path smoothing. In this way, the AGV path can avoid too many sharp turns and unnatural path return, improving the stability of driving.

As shown in Figure 10, the set data are processed using a uniform B-spline curve, and the control points are set to [9.036145, 51.779661], [21.084337, 70.084746], [37.607573, 50.254237], [51.893287, 69.745763]. This set of control point data sets is a randomly generated data set. Its selection has no specific practical meaning but is used to demonstrate the application effect of B-spline curves in path smoothing. As for why the decimal point precision of the control points is high, the reason is that in the B-spline curve optimization process, the algorithm usually uses an iterative method to continuously adjust the position of the control points to improve the accuracy and stability of the path. In order to avoid error accumulation and instability, the algorithm needs to accurately calculate the coordinates of each control point and output a decimal with high precision. In simulation, high-precision control points can reduce errors in path generation, especially when the path passes through complex terrain. The more decimal places the control points have, the higher the accuracy of path planning, which can more accurately reflect actual driving needs. The high precision of the control points ensures that the path remains stable during the optimization process and avoids unnecessary bending or oscillation, improving the smoothness of the path. On the other hand, although these control points do not have actual geographical locations or specific path meanings, they have certain commonalities: the coordinate span between each point is large and distributed in different coordinate regions. This makes the path show higher complexity in the simulation, thereby verifying the effectiveness of B-splines in various path smoothing and optimization tasks. In this way, even a randomly selected data set can still demonstrate the advantages of B-splines in path planning.

The results show that the uniform B-spline image is sinusoidally distributed and equally spaced, making the path smoother and more consistent; when the uniform B-spline is interpolated between a given series of control points, a smooth curve can be generated, and the path can better approach or accurately pass through the target point (control point). Secondly, the uniform B-spline can more stably adapt to the path requirements of different shapes. The B-spline adjusts the shape of the curve by controlling the parameters of the basis function, providing higher flexibility. By selecting appropriate parameters, unnecessary bending or jitter in path planning can be effectively avoided, ensuring the smoothness of the path. This avoids the path distortion and instability problems caused by quasi-uniform B-spline in complex environments [25]. Therefore, this paper uses a close-B-spline curve of the uniform B-spline curve to smoothly optimize the global route.

In order to solve the problems of jagged shape and large turning angle of the global path, this paper adopts the improved B-spline curve to smoothly optimize the global path planning. The specific steps are as follows:

Perform B-spline interpolation through path points. Starting from the original path, set the path point set $P=\{P_{0},P_{1}\ldots P_{n}\}$, and use the B-spline basis function to fit the path points; secondly, approximate the path point set P through the control points $C_{0},C_{1}\ldots C_{n}$, expressed as(9)$$\begin{array}{c} C\left(t\right)=\sum_{i=1}^{m}P_{i}N_{i,3}\left(t\right) \end{array}$$

The expression $N_{i,3}(t)$ represents the cubic B-spline basis function [26], where $P_{i}$ is the control point, and C(t) is the fitted path curve; m is the number of control points that define the curve; t represents a parameter value on an interval; t determines a certain position on the curve; and as t changes, the calculated $C\left(t\right)$ will give the coordinates of the corresponding position.

Based on the set of path points *P* and their distribution, B-splines are used for initial path fitting. The B-spline basis functions generate a smooth curve along the entire path through the control points. The control point $P_{i}$ generates a continuous and smooth curve through the basis function $N_{i,3}(t)$.

For each control point $P_{i}$, local optimization is performed to adjust its position in order to reduce the curvature and unsmooth regions of the curve. In B-splines, the optimization process is achieved by minimizing the cost function. The form of the optimization function is as follows:(10)$$\begin{array}{c} \underset{\left\{P_{1},P_{2},\ldots ,P_{m}\right\}}{min}{t_{0}}_{t_{1}}^{\int }{\left(\frac{d^{2}C(t)}{dt^{2}}\right)}^{2}dt \end{array}$$

Among them, $t_{0}$ represents the start time of path smoothing in the interval, and $t_{1}$ represents the end time of path smoothing. The above formula is used to obtain a smooth fitting curve in the interval [$t_{0}$,$t_{1}$]. $\frac{d^{2}C(t)}{dt^{2}}$ is the second-order derivative of the curve, which represents the curvature of the curve. The optimization goal is to reduce the curvature change and make the curve smoother.

By calculating the ‘importance’ of each control point on the curve, the path can be simplified, and unnecessary control points can be removed. The specific control point importance calculation formula is as follows:(11)$$I_{i}{t_{i-1}}_{t_{i+1}}^{=\int }{\left|\frac{d^{2}C\left(t\right)}{dt^{2}}\right|}^{2}dt$$

$I_{i}$ represents the contribution of the control point $P_{i}$ to the curvature of the surrounding curve segments. If the contribution of the control point is small, the control point is removed.

As shown in Figure 11, the red arrow is the direction of the path starting point, and the green arrow is the direction of the path reaching the end point. After the global path planning realizes the smoothing of the B-spline curve, its movement takes less time and can effectively avoid a wide range of turning angles; for corners with an average turning angle greater than 90 degrees, the turning slope is smoother, the turning point selection path is optimal, and the smoothness is better, which can effectively improve the movement efficiency of the AGV and shorten the path length.

In summary, the overall flow chart of AGV path planning based on the Voronoi skeleton is shown in Figure 12. In this study, the laser radar on the AGV is first used to scan the point cloud data through the Cartography mapping algorithm to generate a cost map. The cost map consists of three levels: obstacle layer, static layer, and expansion layer. Next, the processed Voronoi skeleton image is binarized using the relevant plug-ins in the ROS operating system, and a custom Voronoi layer is generated through iteration. Then, the original three-layer cost map is fused with the custom Voronoi layer to construct a new cost map with a four-layer network structure.

On this basis, the improved A* algorithm is used for global path planning with the new four-layer cost map as a constraint. The planned path has the characteristics of driving along the center of the road, but there is a problem of non-smoothness at the turning point of the path. In order to solve this problem, this study introduces B-spline curve optimization technology to smooth the Voronoi path to obtain the final optimized path.

Next, the feasibility and safety of the proposed algorithm in AGV path planning are verified through simulation and real experimental scenarios.

## 3. Results and Discussion

This paper verifies the feasibility and effectiveness of the path planning algorithm for narrow road scenarios through simulation experiments. The simulation experiment uses the Gazebo simulation platform and the Rviz graphical tool and uses the ROS operating system in the Ubuntu 18.04 system for simulation. Among them, Gazebo is used to build a 3D model of the simulation environment, including the AGV and its surroundings, and Rviz is used to display the AGV motion status and navigation path in real time. The simulation experiment uses a wheeled differential AGV model with a square chassis and a two-dimensional laser radar on the front and back.

In the simulation environment of this study, the AGV size parameters are as follows: the chassis size is 0.266 m × 0.266 m × 0.094 m; the wheel radius is 0.033 m, and the length is 0.018 m; the radius of the laser sensor is 0.055 m, and the length is 0.0315 m. The shape of the simulation map is approximately hexagonal, with a total size of 5 m × 5 m, the radius of the obstacle is 0.15 m, and the path width between the two obstacle pillars is set to 0.52 m, which meets the definition of narrow road width; this map is used to simulate narrow road environment scenarios. These dimensions provide physical constraints for the movement and path planning of AGV in the simulation environment, ensuring the authenticity and effectiveness of the experiment. By setting the path width and obstacle size, the study can verify the navigation and obstacle avoidance capabilities of AGV in limited space and complex environments, especially in the application of path optimization and obstacle avoidance strategies.

### 3.1. Verify the Security of the Algorithm in a Simulated Environment

As shown in Figure 13a, the A* algorithm ignores the volume of the AGV when planning the path, resulting in the planned path being close to obstacles, especially when passing through narrow passages, where the AGV is prone to collision when in contact with obstacles. As shown in Figure 13b, this is the global path planning result of the Voronoi layer algorithm proposed in this paper. Compared with the A* algorithm, the global path planning of this algorithm introduces the Voronoi layer to optimize the path selection, ensuring that the AGV travels along the center line of the safe area, effectively avoiding the planned path from being close to obstacles, improving the efficiency and stability of path planning, and greatly reducing the computational complexity of local path planning in narrow passages.

This paper uses the new cost map of the four-layer network structure as a constraint and uses the improved A* algorithm for depth-first search to generate a global path with high security and efficiency. As shown in Figure 14a, the path is kept on the center line of the road to the greatest extent to stay away from obstacles. However, at the turning point of the path, the line shows a large tortuosity, and a large turning angle will cause the AGV to travel unsteadily; in local path planning, the frequent turning of the AGV will affect the stability of the path tracking.

In view of the instability and real-time performance of AGV navigation caused by the above global path planning, this paper uses B-spline curves based on the Voronoi layer algorithm to smoothly optimize the path at the turning point. Although the path planning of the Voronoi algorithm has greatly improved in terms of safety distance, it is prone to problems such as excessive turning angles and too many turning times when facing the situation of inconsistent front and rear widths of narrow passages. As shown in Figure 14b, the global planning path optimized by the B-spline curve has greatly improved the smoothness of the turning points of the path, reducing the jamming phenomenon and the number of turnings when the AGV moves and reducing the complexity of local path calculation and movement delay.

Special note: The red lines in the following figure represent the Voronoi skeleton in the customized Voronoi layer, which is used to describe the relationship between boundaries and obstacles in path planning. It serves as auxiliary information for reference paths or navigation. The appearance of red lines intuitively shows the critical path in the Voronoi diagram. The red lines in Figure 14a,b have the same meaning, representing the Voronoi skeleton in the customized Voronoi layer.

This paper builds a simulation scene of a narrow road environment and uses the path planning of this algorithm to compare and analyze other algorithms. As shown in Figure 15a, this is the global path planning performed by the A* algorithm in the cost map simulation environment. Although the path can search for the optimal route, it uses a heuristic depth-first search, which makes the path too close to obstacles and prone to collisions. Therefore, it is difficult for the A* algorithm to effectively meet the safety and reliability requirements of narrow road scenes.

As shown in Figure 15b, compared with the A* algorithm, the global path planning of the Voronoi algorithm proposed in this paper is closer to the center line of the road and significantly away from obstacles, which can improve the safety of the path. However, in the narrow road scene, the turning position has a large turning angle and too many turning times, resulting in low path planning efficiency and large delay during AGV movement.

As shown in Figure 15c, this paper uses B-spline curves to smoothly optimize the global path generated by the Voronoi algorithm. Compared with Figure 15b, its path is smoother, which makes the turning angle smaller and the number of turning times significantly reduced and reduces the computational complexity and movement delay while ensuring safety. Therefore, the improved algorithm is particularly suitable for path planning in complex scenarios such as narrow roads.

As shown in Table 1, compared with the traditional A* algorithm, in narrow road scenarios, the path lengths of the Voronoi layer algorithm and the proposed algorithm increased by 0.26% and decreased by 4.04%, respectively; the number of turning points increased by 75% and 25%, respectively; the average distances to obstacles increased by 100% and 75%, respectively; and the planning time decreased by 31.9% and 17.34%, respectively.

Simulation experiments show that compared with the traditional A* algorithm, the path of this algorithm in narrow road scenes is significantly safer, and compared with the Voronoi layer algorithm, it has better path smoothness and can avoid obstacles to the greatest extent. Simulation experiments show that the risk of obstacle avoidance in this algorithm is reduced by 75%, which improves the ability of AGV to adapt to the environment.

### 3.2. Analysis of the Time and Space Complexity of This Algorithm

In this section, we will analyze the time complexity and space complexity of the two core parts of the proposed algorithm, the Voronoi algorithm and the B-spline optimization in detail. The analysis results will help evaluate the computational efficiency and scalability of the algorithm in different scale environments and ensure its feasibility in practical applications.

The construction of the Voronoi diagram is a key step in the algorithm, and its time complexity mainly depends on the number of seed points in the diagram and the algorithm for constructing the diagram.

Time complexity: For the construction of a Voronoi diagram with n seed points, the time complexity of the Voronoi algorithm is O (n log n), where n is the number of seed points in the diagram. Specifically, the construction process includes the processing of various events and the maintenance of related data structures. The time complexity of these operations is mainly O (log n) queue operations, resulting in an overall time complexity of O (n log n).

Influencing factors: The time complexity of constructing the Voronoi diagram is greatly affected by the number of seed points n. As the number of obstacles in the environment increases, the number of points will also increase significantly, resulting in an increase in construction time. Therefore, the efficiency of Voronoi diagram construction is particularly important for large-scale environments.

B-spline optimization plays an important role in path smoothing. This process generates a smooth path by interpolating and optimizing the control points. The B-spline optimization process mainly involves calculating the influence of each control point on the curve shape and adjusting the position of the control points according to the optimization algorithm.

Time complexity: The time complexity of B-spline optimization is O(m), where m is the number of control points. The calculation of B-spline is essentially an interpolation calculation and weighted average of each control point, so its time complexity is linearly related to the number of control points. During the optimization process, we need to adjust the control points according to the algorithm.

Influencing factors: The time complexity of B-spline is closely related to the smoothness requirement of the path. If the path requires more control points to provide a finer smoothing effect, the number of control points m will increase, resulting in an increase in calculation time. In general, the time cost of B-spline optimization is relatively low, but as the number of control points increases, the time cost of optimization calculation may increase significantly.

In the space complexity analysis, we will focus on the storage requirements of the algorithm in memory. Specifically, we will analyze the storage requirements of the Voronoi diagram and the storage requirements of the B-spline control points.

Space complexity of Voronoi diagram: Voronoi diagram needs to store all nodes and their connections in the graph, which usually requires, $O\left(n\right)$ space, where n is the number of seed points in the graph. In addition, the construction process may require the use of additional data structures to store the edges, faces, and event queues of the graph, resulting in a space complexity of $O\left(n\right)$.

Space complexity of B-spline optimization: The space complexity of B-spline is mainly determined by the number of control points m to be stored. In order to generate a smooth path, B-spline needs to store control points and their associated parameters, which leads to a space complexity of O(m). In addition, temporary variables may be used for calculations during B-spline optimization, but their additional space requirements are relatively small.

To sum up, the time complexity and space complexity formulas of this algorithm are as follows:(12)$$\begin{array}{c} \left\{\begin{array}{c} T=O\left(nlogn\right)+O\left(m\right)\\ S=O\left(n\right)+O\left(m\right) \end{array}\right. \end{array}$$

T represents time complexity; S represents space complexity; *n* represents the number of seed nodes in the Voronoi diagram; *m* represents the number of control points in the B-spline optimization process.

In practical applications, as the scale of the environment increases, the time complexity of the Voronoi algorithm may become a key factor limiting the performance of the algorithm. Especially in complex environments, the increase in the number of obstacles will lead to a significant increase in the number of points in the graph, thus affecting the construction speed. Therefore, in order to improve the scalability of the algorithm, it is possible to consider using a more efficient Voronoi diagram construction algorithm or segmenting and parallelizing the graph in a large-scale environment. In addition, the space and time overhead of B-spline optimization is relatively small and usually does not become a bottleneck, but in scenarios where a fine and smooth path is required, the increase in the number of control points may still affect the efficiency of the algorithm.

### 3.3. Experimental Verification and Analysis of Real Scenarios

In order to verify the correctness and effectiveness of the algorithm in this paper in real scenarios, this paper uses a self-developed two-wheel differential mobile AGV for path planning experiments and analysis. The AGV measures 70 cm × 50 cm × 25 cm; is equipped with a single-line laser radar, a gyroscope, and an accelerometer; and is equipped with an RK3588 control board and an infrared sensor. As shown in Figure 16, an AGV test vehicle for real-scene testing is shown. The vehicle is equipped with multiple key sensors and functional modules to ensure its efficient operation in complex environments. A laser radar is mounted on the roof for environmental perception and obstacle detection; infrared sensors are used to assist in positioning and collision avoidance. In addition, the vehicle is also equipped with an emergency brake button that can quickly stop the vehicle in an emergency to ensure safe operation. The contact charging port facilitates automatic charging of the AGV during autonomous driving to ensure its continuous operation capability. The front end of the vehicle body is equipped with an LED display to display the device status and operation information in real time and provide intuitive operation feedback.

The experimental site tested in this paper is an irregular narrow environment, similar to a long corridor, with an elevator corridor on one side and several rooms on the other side. The maximum length of the entire scene in the north–south direction is 120 m, and the maximum width in the east–west direction is 50 m. As shown in Figure 17, the figure shows the grid cost map generated by the laser scanning mapping algorithm, which clearly shows the walls, obstacles, and corridor layout of the environment. The map is irregular in shape, with obvious boundaries between corridors and rooms, and obstacles distributed in different locations. The unknown area has a higher cost value and is marked in gray, indicating that it is not passable; while the open area has a lower cost value, indicating that it is passable. This cost map provides accurate environmental information for AGV path planning.

As shown in Figure 18, in a narrow road scenario, when the A* algorithm is used for path planning, the red line represents the planned path from start point 1 to end points 2 and 3, and the green line represents the return path. The A* algorithm only considers the shortest path, resulting in the planned route being close to the obstacle wall, failing to fully consider the volume constraints of the AGV itself, lacking consideration for the safe driving space of the AGV, resulting in poor safety, and failing to optimize operability in the actual driving environment.

In response to the above problems, the algorithm in this paper extracts Voronoi skeleton features and integrates them into the cost map, iteratively generates a new cost map with a four-layer network structure, and then introduces B-spline curves to locally optimize the path after searching the global path with A* heuristic path, reducing sharp turns and irregular turns in the path, so that the AGV can drive safely close to the center in narrow road scenes. Secondly, adding the optimized path with B-spline curves reduces the turning in the movement of the AGV, thereby effectively improving the smoothness of the path. Finally, comparing the two path planning lines generated by the A* algorithm in Figure 18, as shown in Figure 19, when the map scene, starting point, end point, and other variable conditions are consistent, the algorithm in this paper only generates one path planning line for the reciprocating motion of the AGV, which greatly reduces the repeated path points and effectively improves the quality of path planning and the efficiency of AGV planning. In the actual vehicle test, the planned routes of the two algorithms are compared. The total path length planned by the algorithm in this paper is shorter and the path planning quality is better.

As shown in Figure 20 and Figure 21, the narrow road scene was partially enlarged by Rviz 1.10 visualization software, and the path generated by the Voronoi layer algorithm was compared with the path optimized by the B-spline curve in both local and overall aspects. As shown in Figure 20a,b, the comparison results show that the path optimized by the B-spline curve has significant improvements in global smoothness and greatly improved driving stability. In actual tests, the AVG optimized by the B-spline curve showed a smoother driving state when turning, avoiding the phenomenon of frequent turning and changing direction during movement. As shown in Figure 21a,b, the comparison between the 1 and 2 marks in the figure clearly shows that the path after B-spline optimization is smoother, while the unoptimized path has more jagged shapes.

As shown in Table 2, according to the average value of the experimental test data, the Voronoi layer algorithm and the proposed algorithm reduced the path length by 5.4% and 7.8%, respectively; the average distance from obstacles increased by 73% and 82%, respectively, greatly improving the safety of AGV movement; in terms of the number of points in the path planning space, the two algorithms reduced by 53.19% and 55.85%, respectively, and the planning time was reduced by 29.07% and 48.98%, respectively, thereby reducing the amount of calculation of the system. The above results show that the algorithm proposed in this paper has better operating efficiency, stability, and reliability in experimental tests.

In order to verify the performance of the algorithm in different narrow road environments, this study designed multiple experimental scenarios with different widths. Specifically, three different road widths were simulated in the experiment, namely, 1.44 m, 3.12 m, and 0.82 m, to simulate different degrees of narrow road environments. The above road widths are taken from real scenes: the narrow road scene from the room to the corridor, the planned scene in the elevator corridor, and the scene in the middle narrow corridor, corresponding to the scene diagrams of Figure 22a–c, respectively. The experimental data are shown in Figure 23. The horizontal axis represents the execution time of the algorithm under three narrow roads of different widths, and the vertical axis represents the distance from the AGV to the end point. The smoothness of the path at different widths is evaluated by drawing the curve shape of the path. The smoothing formula is as follows:(13)$$\begin{array}{c} k=\frac{\left|a\;\nu \right|}{\nu^{3}} \end{array}$$

*κ* is the curvature; a is the acceleration; *v* is the velocity; this formula reflects the geometric characteristics of the curve in plane motion. Curvature represents the degree of curvature of the trajectory, |a ν| represents the absolute value of the product of acceleration and velocity, reflecting the curvature of the trajectory, and $\nu^{3}$ represents the degree of curve change in motion through the cube of velocity.

The change in path smoothness can be clearly seen in the figure. For a road with a width of 3.12 m, the path curve changes smoothly, the path is smooth, and the time taken is the shortest. When the width is 1.44 m, the path shows a relatively smooth curve, without any drastic turns or fluctuations overall. When the width is 0.82 m, although the road is narrow, the algorithm still maintains good smoothness and avoids large turns, demonstrating the stability of the algorithm under extreme conditions.

According to Table 3, we can conclude that although the calculation time of the proposed algorithm is extended on roads of different widths, its smoothness is still effectively guaranteed. The specific analysis is as follows: for road widths of 1.44 m, 3.12 m, and 0.82 m, the curvatures correspond to 0.34412, 0.20623, and 0.42589, respectively. Although the narrower the width, the greater the curvature, in actual tests, the experiments on the extremely narrow 0.82 m, the route smoothness obtained can still meet the planning requirements, and the path is relatively smooth.

## 4. Conclusions

In view of the problems of insufficient safety distance, uneven path, low planning efficiency, etc., in AGV path planning in narrow road scenarios, this paper proposes a narrow road path planning method based on the Voronoi skeleton. The main research conclusions are as follows:

This paper extracts the key points of the Voronoi skeleton to generate a custom Voronoi layer; integrates the static layer, obstacle layer, and extension layer; and iteratively generates a new cost map with a four-layer network structure.

The experimental results show that the path planning safety index of the proposed algorithm is improved by 82%, which effectively improves the safety distance during the operation of AGV and ensures the safe driving of AGV in complex environments. By optimizing the path planning, the proposed algorithm can more accurately maintain and adjust the safety distance, especially in narrow roads and environments with dense obstacles, effectively reducing the risk of collision and further improving the safety and stability of AGV.

In view of the problems of jagged paths and many turning points in narrow road scenarios, the B-spline curve smoothing optimization algorithm is introduced. The generated smooth path guides the AGV to perform local path planning, realizes the smoothness optimization of the Voronoi layer algorithm, reduces the amount of spatial control data, and improves the stability of the AGV driving path. The experimental results show that the execution time of path planning is shortened by 48.98%, and the number of planning space points is reduced by 55.85%, which effectively improves the driving stability of AGV in narrow environments.

This paper verifies the algorithm through simulation and real environment experiments, respectively. The experimental results verify the safety, robustness, and efficiency of the fusion path planning method in this paper, which can quickly avoid obstacles and plan the global optimal path and improve the environmental adaptability of AGV in narrow road scenes.

However, the path planning algorithm proposed in this paper mainly focuses on the problem of avoiding static obstacles and has not yet involved the processing of dynamic obstacles (such as moving obstacles), which is a limitation of this study. Future research will focus on solving the problem of avoiding dynamic obstacles and further enhance the adaptability and practicality of the algorithm. With the increase in the complexity of dynamic environments, path planning considering dynamic obstacles will become an important direction for future research.

Potential enhancements include the following:

Dynamic obstacle avoidance: Introduce real-time sensor data, focus on modifying the local path planning strategy, and adjust the clearing strategy of the dynamic obstacle cost map to ensure that the AGV avoids obstacles in real time in a dynamic environment.

Through RL (reinforcement learning), AGV gradually optimizes the path planning strategy through interaction with the environment. AGV will try and err in various situations and gradually learn the optimal obstacle avoidance strategy. Especially when there are many dynamic obstacles, the RL model will help AGV continuously adjust the strategy to obtain the optimal path and avoid collisions. On the other hand, the reinforcement learning model is used to learn to set avoidance priorities, such as giving priority to avoiding quickly approaching obstacles or determining obstacle avoidance strategies based on factors such as the type and speed of the obstacle.

## Figures and Tables

**Figure 1 sensors-25-02786-f001:**
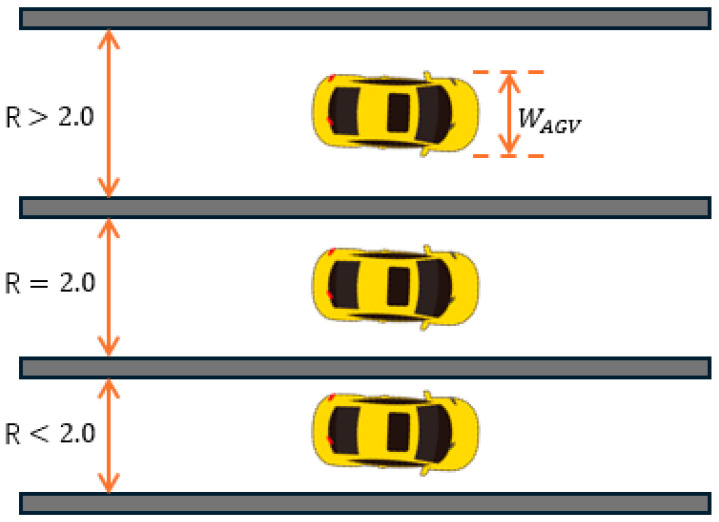
This diagram shows the definition of a narrow road scenario. There are three roads in the figure. When the ratio of road width to AGV is less than or equal to 2.0, for example, the second and third roads in the figure are defined as narrow road scenarios.

**Figure 2 sensors-25-02786-f002:**
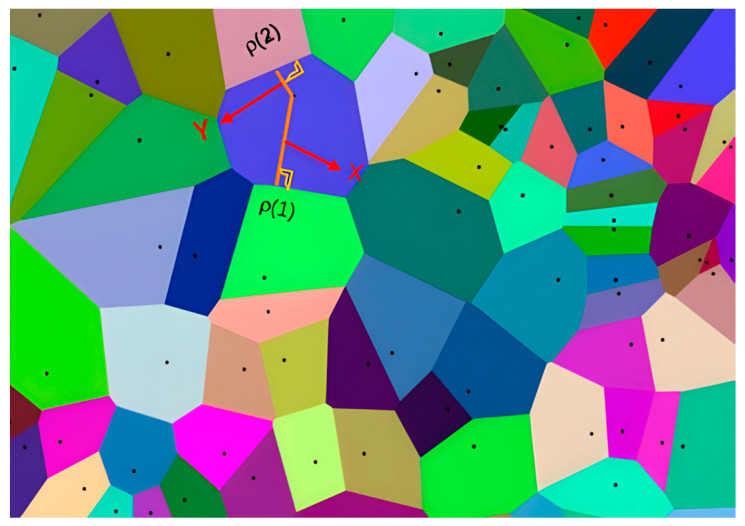
This figure is a schematic diagram of the seed point area of the Voronoi diagram, where each polygon represents a Voronoi unit, and each point in the unit is closest to the generating seed point of the unit.

**Figure 3 sensors-25-02786-f003:**
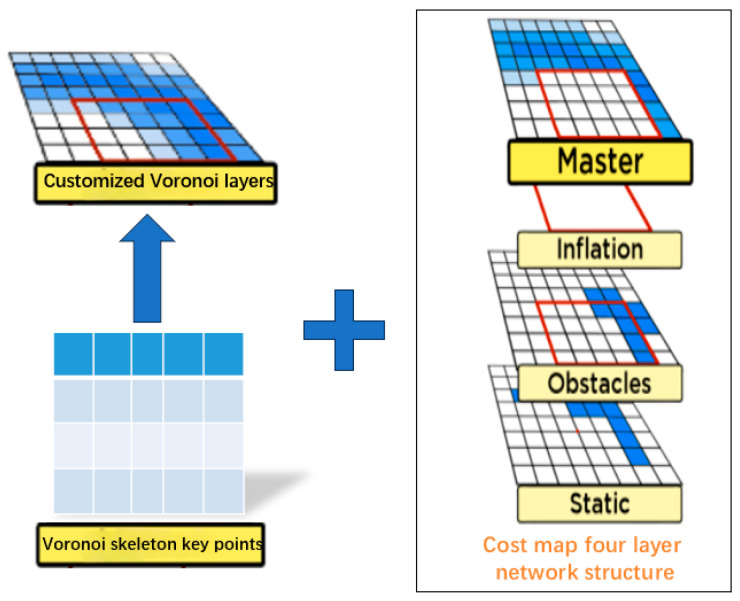
This image presents a cost map based on a four-layer network structure. On the left side of the image, the key points of the Voronoi skeleton are highlighted. The arrow on the left indicates the Voronoi layer processed through binarization and thinning. The plus sign on the right illustrates the four-layer network structure formed by combination, which includes the master layer, inflation layer, obstacle layer, and static layer. This structure is designed to optimize path planning in narrow road scenarios. Among them, the white squares represent unknown safe or areas, and the blue squares represent obstacle areas.

**Figure 4 sensors-25-02786-f004:**
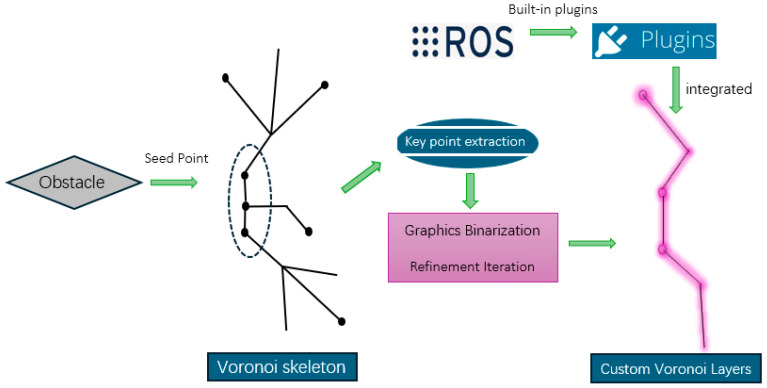
This figure shows the process of Voronoi skeleton extraction and custom Voronoi layer generation: obstacles are used as seed points to generate Voronoi skeletons through the Voronoi algorithm; key point extraction and graphic binarization are used to simplify the Voronoi skeleton into important path points; redundant points and line segments are removed through refinement iteration to optimize the skeleton structure; and the generated Voronoi layer is integrated into the system using a ROS plugin.

**Figure 5 sensors-25-02786-f005:**
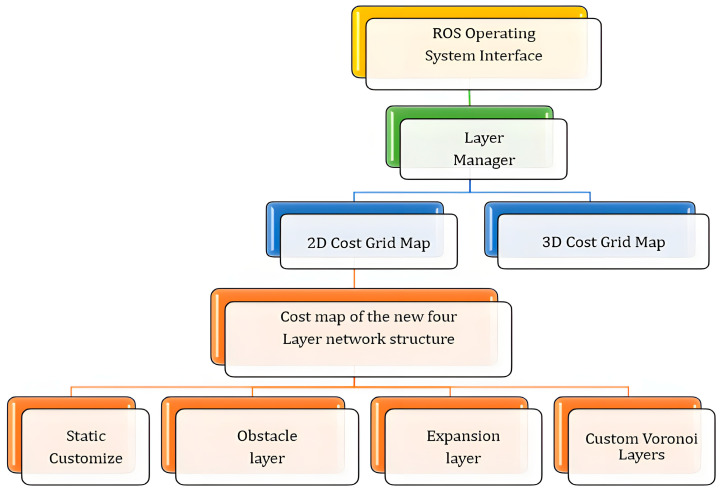
This image illustrates a four-layer network cost map structure based on ROS. At the top, the ROS operating system interface and layer manager are responsible for managing map layers. The middle section consists of 2D and 3D cost grid maps, both supporting path planning. At the bottom, the four-layer cost map includes the static layer, customized Voronoi layer, and expansive layer.

**Figure 6 sensors-25-02786-f006:**
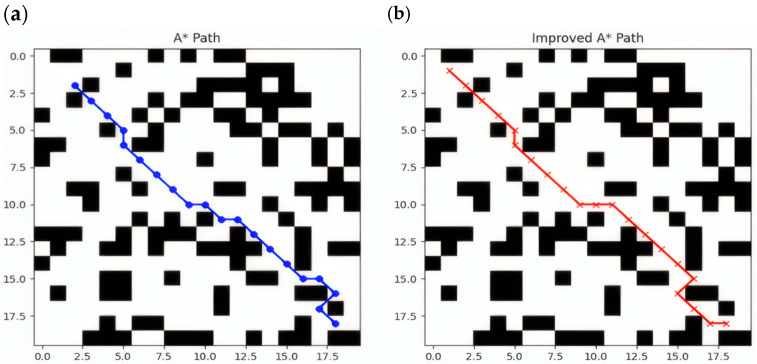
This figure represents a comparison chart of the paths generated by the A* algorithm and the improved A* algorithm. In the figure, the blue line segments in (**a**) represent the path generated by the A* algorithm when performing path planning in this complex and narrow scenario. The red line segments in (**b**) of the figure represent the path generated by the improved A* algorithm when conducting path planning in the same complex and narrow scenario.

**Figure 7 sensors-25-02786-f007:**
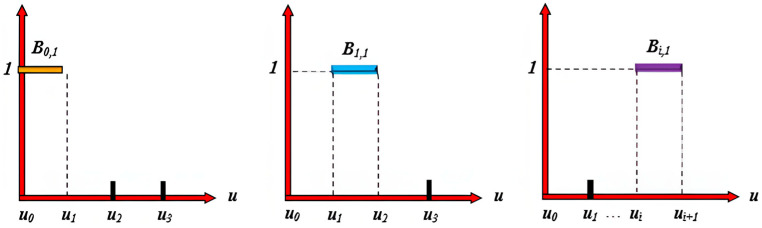
The image illustrates the shape of the B-spline basis function $B_{i,1}(u)$. Each subfigure represents the basis function under a different index i, with its domain determined by the knot vector ${[u}_{0},u_{1},u_{2},\ldots ,u_{i+1}]$. The basis function takes a value of 1 within the corresponding interval and 0 elsewhere, forming a step-like distribution.

**Figure 8 sensors-25-02786-f008:**
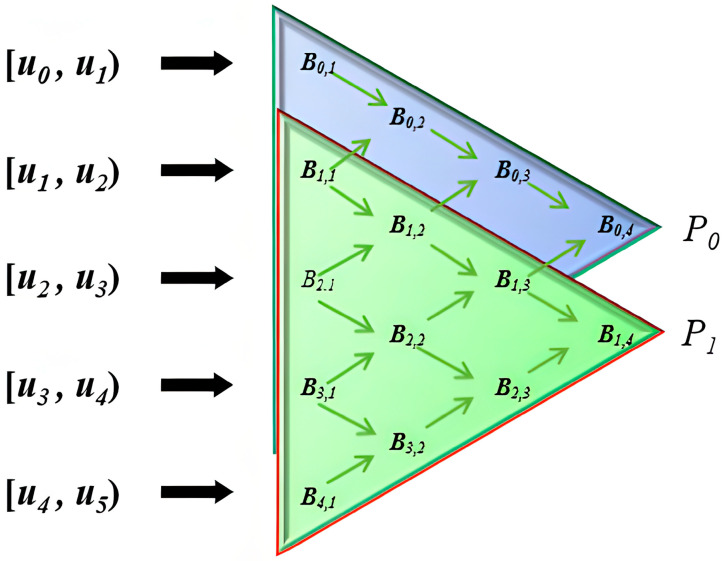
The diagram represents the recursive computation process of B-spline basis functions, illustrating the recursive relationships across different parameter intervals. The arrows indicate the direction of weight propagation, computing control points $B_{i,k}\left(u\right)$ step by step to ultimately generate a smooth curve or surface.

**Figure 9 sensors-25-02786-f009:**
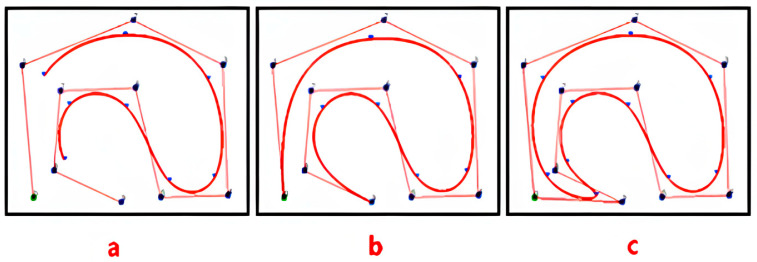
(**a**) represents an open B-spline curve, with fixed start and end points, that is not closed. (**b**) represents a semi-open B-spline curve, with one end fixed and semi-closed. (**c**) represents a closed B-spline curve, a fully closed curve.

**Figure 10 sensors-25-02786-f010:**
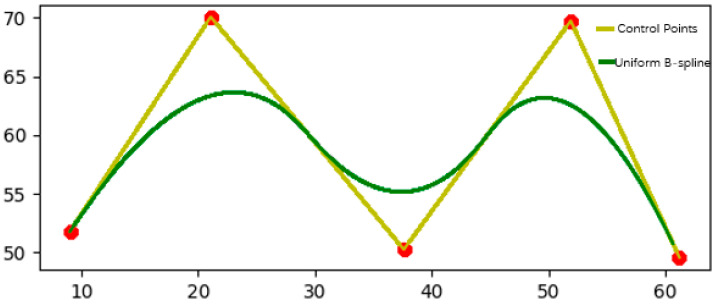
The yellow line in the figure represents the control points, the red dots represent the set points, and the green line represents the uniform B-spline fitting smoothing of the data points.

**Figure 11 sensors-25-02786-f011:**
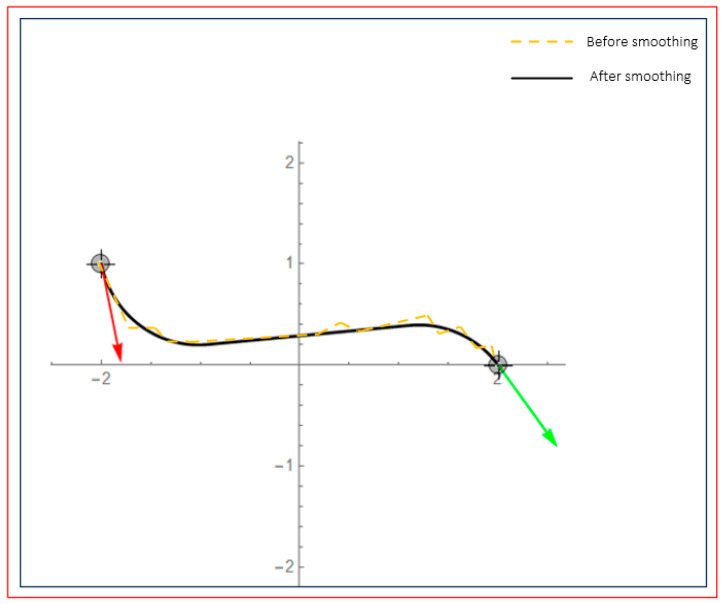
This image shows the path planning effect before and after smoothing. The red curve represents the path before smoothing, while the black curve represents the path after smoothing. The red and green arrows in the image represent the starting and ending directions of the path, respectively.

**Figure 12 sensors-25-02786-f012:**
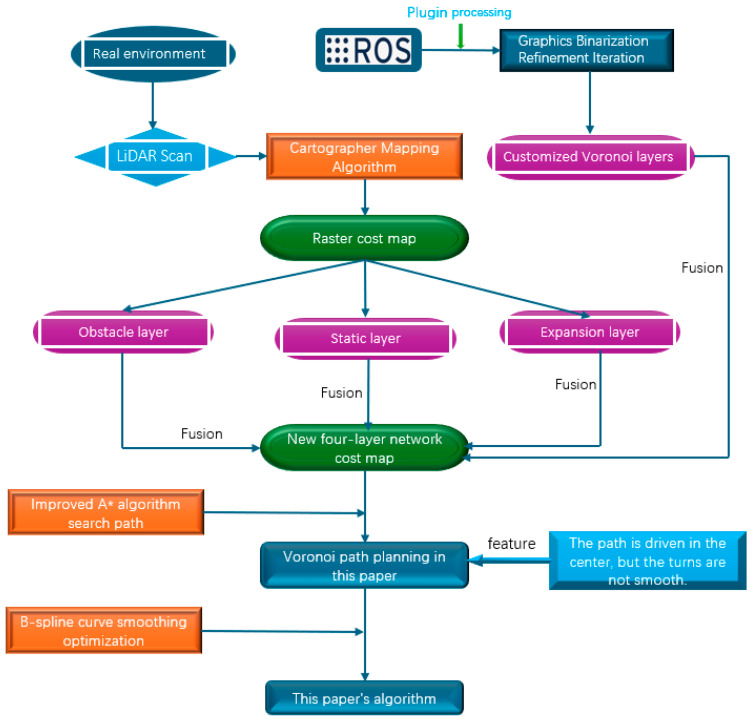
The figure shows the overall process of AGV path planning based on the Voronoi skeleton. The process includes scanning point cloud data to generate cost maps of obstacle layer, static layer, and expansion layer; binarizing the Voronoi skeleton to generate a custom Voronoi layer; then fusing the Voronoi layer with the three layers of the original cost map to obtain a cost map with a four-layer network structure; using the improved A* algorithm for path planning; and using the B-spline curve to smoothly optimize the path to obtain the final planned path.

**Figure 13 sensors-25-02786-f013:**
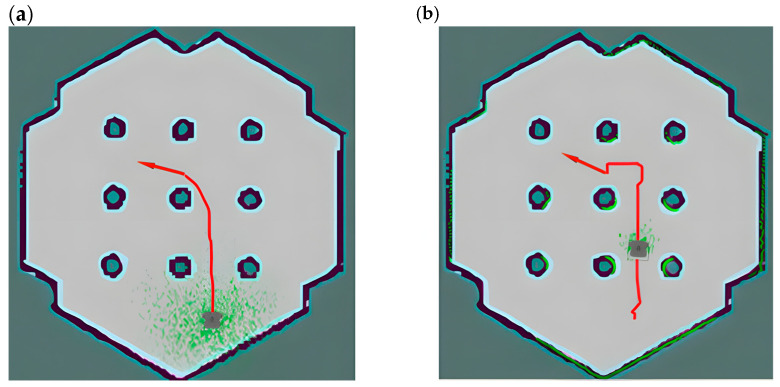
The figure is a comparison between the A* algorithm and the path planning algorithm in this paper. (**a**) shows the path planning of the A* algorithm, and (**b**) shows the path planning of the improved A* algorithm combined with the four-layer network structure cost map in this paper, the red line segments in the figure represent the paths planned using different algorithms, and the dotted green area represents a schematic diagram of the radar point cloud.

**Figure 14 sensors-25-02786-f014:**
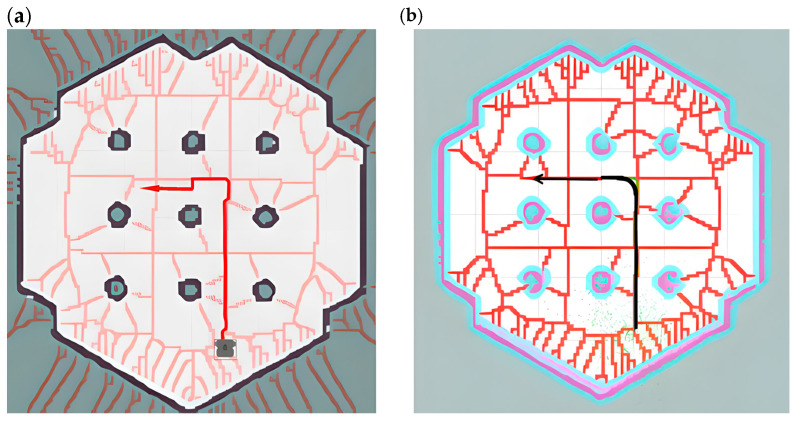
The figure shows a schematic diagram of the path comparison before and after B-spline optimization, where (**a**) shows the path planning roadmap constrained by the cost map of the four-layer network structure proposed in this paper, and (**b**) shows the path planning roadmap after B-spline optimization based on the above algorithm, the arrows in the figure indicate the direction of the destination, which determines the final orientation of the vehicle when it reaches the destination.

**Figure 15 sensors-25-02786-f015:**
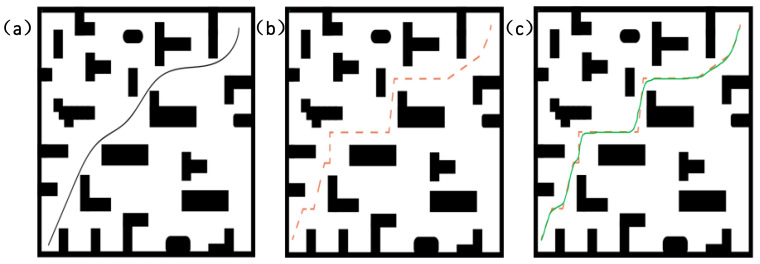
This figure shows the comparison of the paths planned by three algorithms in a narrow road scenario. (**a**) shows the path planning route of the A* algorithm, (**b**) shows the path planning diagram using the cost map constraint of the four-layer network structure, and (**c**) shows the path planning diagram using B-spline optimization.

**Figure 16 sensors-25-02786-f016:**
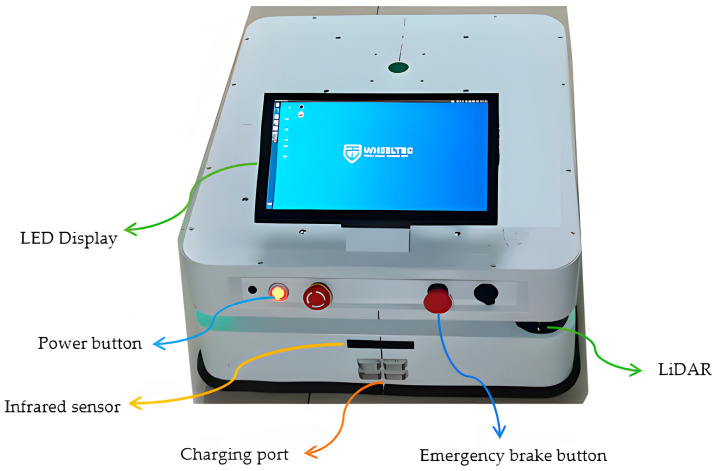
This figure shows the hardware composition and functional module diagram of the AGV test vehicle. The figure shows the main hardware configuration of the AGV test vehicle, including laser radar, infrared sensor, emergency brake button, contact charging port and LED display.

**Figure 17 sensors-25-02786-f017:**
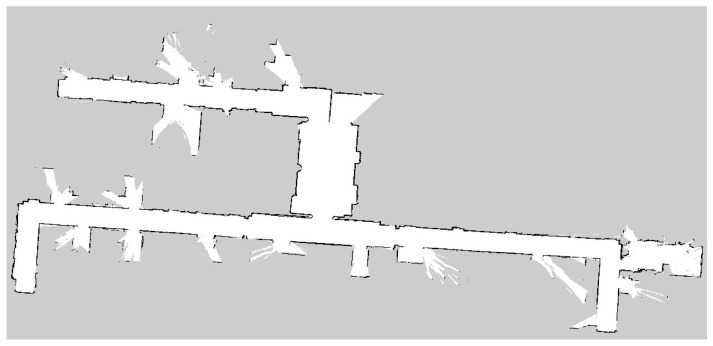
This image is a raster cost map generated by laser scanning, which clearly shows the distribution of walls, obstacles, and traversable areas.

**Figure 18 sensors-25-02786-f018:**
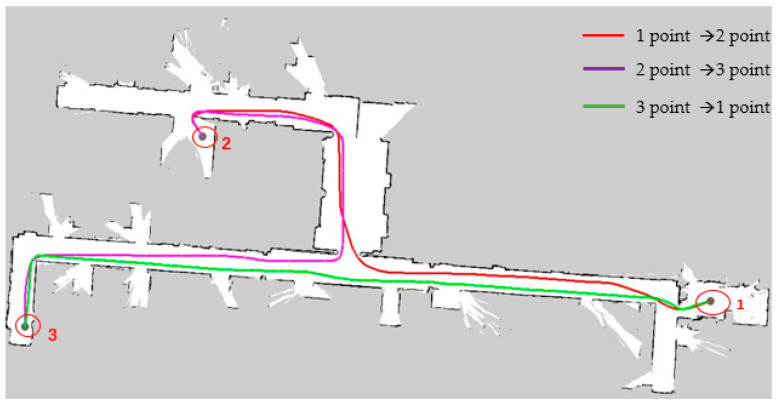
This figure shows the path planning of the A* algorithm in a narrow experimental scene in the Rviz window, where the red line segment represents the path from point 1 to point 2, the purple line segment represents the path from point 2 to point 3, and the green line segment represents the path from point 3 back to point 1. In the figure, 1 and 3 represent the starting point and the end point, and point 2 represents the transition point.

**Figure 19 sensors-25-02786-f019:**
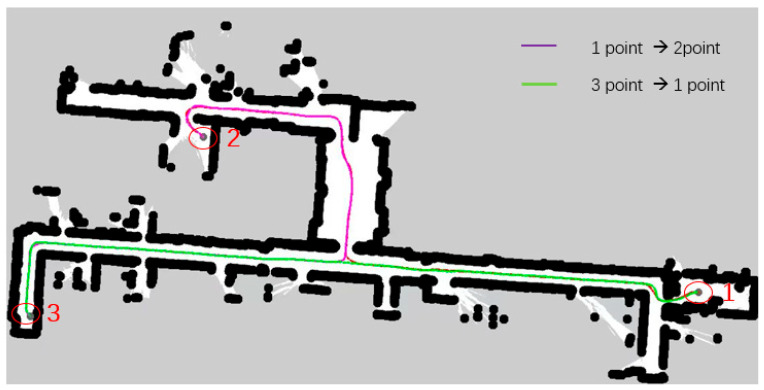
The following figure shows the path planning of the algorithm in a narrow road scenario. The purple line segment represents the route from starting point 1 to transition point 2. A part of it is covered by the green line segment, so only the purple line segment after the fork is clearly observed. The green line segment is the path segment from end point 3 back to starting point 1. The black dots in the figure represent the path boundaries.

**Figure 20 sensors-25-02786-f020:**
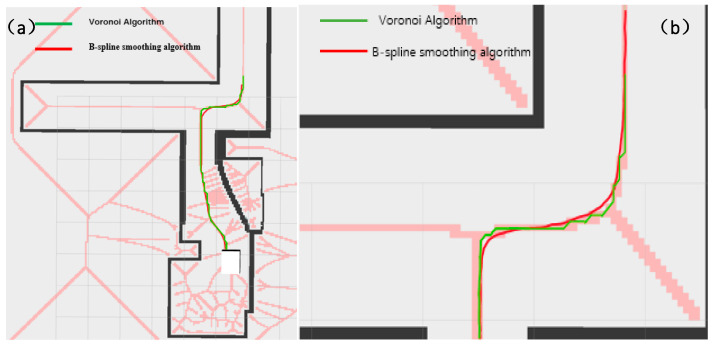
This figure shows the local comparison effect of the Voronoi layer algorithm and the B-spline optimization smooth path. (**a**) shows the comparison of the overall path, and (**b**) shows the comparison of the local magnification effect. The green represents the Voronoi layer algorithm path, and the red represents the path after B-spline optimization. The pink line in the figure represents the constrained area generated by the Voronoi algorithm. The white square represents the test vehicle model, and the black line segment represents the map boundary.

**Figure 21 sensors-25-02786-f021:**
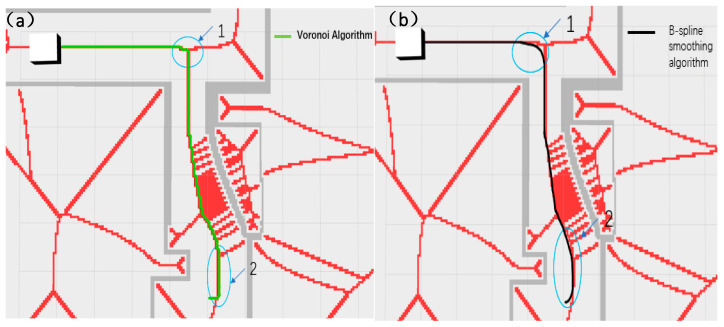
This figure shows the comparison of the Voronoi layer algorithm proposed in this paper and the algorithm route after B-spline optimization when the AGV passes through a narrow door frame. (**a**) shows the path effect diagram of the Voronoi algorithm, and (**b**) shows the path effect diagram after B-spline optimization. The two areas 1 and 2 in (**a**,**b**) indicate the obvious differences between the two algorithms when turning. Similar to the previous figure, the red lines in this figure represent the constrained areas generated by the Voronoi algorithm, and the white squares represent the test vehicle models.

**Figure 22 sensors-25-02786-f022:**
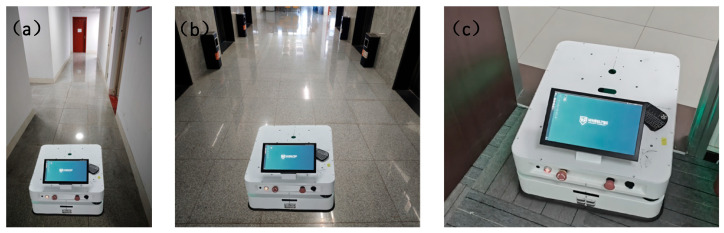
This figure shows the real experimental scene of narrow roads of different widths. The three scene pictures are taken from the door frame scene, narrow corridor scene, and elevator scene test in the real environment to ensure the uniformity of the algorithm and the authenticity of the narrow road width. (**a**) shows the testing of the vehicle’s performance in a road environment with a width of 0.82 m. (**b**) represents the testing carried out within a width of 1.44 m. (**c**) depicts the testing of the vehicle’s performance in a road environment with a width of 3.12 m.

**Figure 23 sensors-25-02786-f023:**
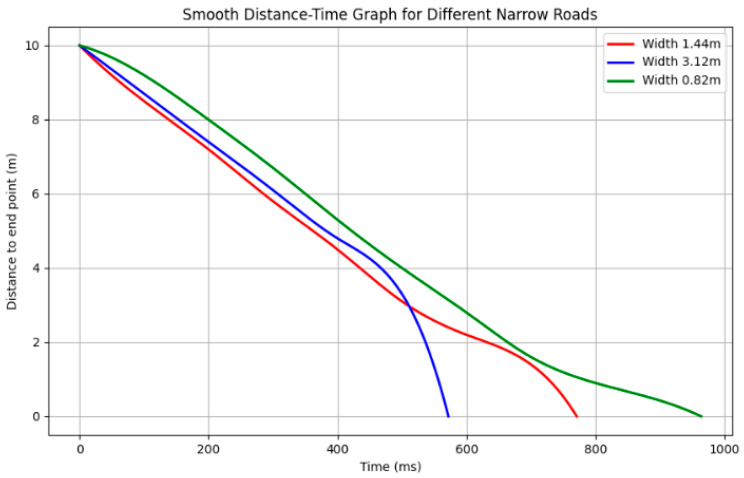
This figure shows a comparison of the path smoothing effects of the algorithm under different widths. The horizontal axis is the length of the algorithm time, and the vertical axis is the distance to the set point. It can be seen from the figure that as the road becomes narrower, although the execution time increases slightly, the smoothness of the path can still be well maintained, indicating that the algorithm has strong adaptability.

**Table 1 sensors-25-02786-t001:** Comparison of simulation performance of narrow road path planning.

Algorithm	Path Length (m)	Number of Turns	Distance to Obstacles (m)	Planning Time (ms)
A* Algorithm	19.5479	4	0.21	15.34
Voronoi Layer	19.5998	7	0.4	10.445
Proposed Algorithm	18.7580	5	0.35	12.662

**Table 2 sensors-25-02786-t002:** Experimental performance comparison of path planning in a real narrow road scenario.

Algorithm	Path Length (m)	Distance to Obstacles (m)	Path Planning Space Points	Planning Time (ms)
A* Algorithm	203	0.23	245,663	2466
Voronoi Layer	192	0.41	114,981	1749
Proposed Algorithm	187	0.39	108,447	1258

**Table 3 sensors-25-02786-t003:** Performance comparison of path planning algorithms under different road widths.

Road Width (m)	Computation Time (ms)	Set Distance(m)	Curvature Changes
1.44	771	10	0.34412
3.12	572	10	0.20623
0.82	964	10	0.42589

## Data Availability

Data are contained within this article.

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
