# Peer review of "Path Planning in Narrow Road Scenarios Based on Four-Layer Network Cost Structure Map"

_sensors, 2025, doi:10.3390/s25092786_

Round 1
Reviewer 1 Report
Comments and Suggestions for Authors
The authors propose a novel path planning approach based on Voronoi diagrams and B-spline smoothing. The paper looks well-written and easy-to-follow, authors’ ideas are clear, but some important information and details are omitted, making it challenging to reproduce the developed techniques and obtained results. Please address the comments below.
Major comments:
- In the paper, the authors analyze path planning in narrow road scenarios. The authors, however, do not explain what they actually mean under word “narrow.” Does it mean a specific ratio between the AGV dimensions and road width? I recommend the authors give more details.
- As a consequence to the comment above, it becomes unclear where this “narrowness” is addressed in the proposed algorithms. At the moment, authors’ techniques look applicable to any scenarios, and the emphasis on narrow roads is not observed in the theoretical background (Section 2).
- Section 2.1. It is unclear (at least to me) how the authors actually derive the Voronoi diagram from the obstacles positions. What is the input data for computing this diagram and how do the authors select the seed points? I recommend the authors provide more details about their approach, which constitutes the crux of the paper.
- Section 2.2. It is also unclear how the authors extract key points from the Voronoi skeleton and generate a customized Voronoi layer. This section describes the neural network but do not give any specific details, which could be important for reproducing authors’ methods. If possible, I recommend the authors give more details about the developed network.
- Section 2.3. Similar to the previous comments, this section also lacks details. For example, it is unclear how the authors obtained the results presented in Fig. 4: the authors do not specify any numerical values and simulation parameters. The theoretical foundations look poorly explained too. For example, in l. 185, the authors mention a turning cost, but do not provide any expressions. Thus, it becomes challenging to implement authors’ techniques and reproduce the obtained results.
- P. 8, l. 259:
- It is unclear what the authors mean under the “stability characteristics of the B-splines.”
- The authors write, “the closed B-spline curve has the best smoothing performance,” but it is unclear how to use this curve if the planned path is not closed.
- P. 8, l. 267–268. How did the authors deduce the values of the control points? Is it just a random example or do these values correspond to any specific path planning problem?
- Section 3 does not specify simulation parameters, including AGV dimensions, road dimensions, parameters of the A* algorithm and the improved A* algorithm, neural network parameters, and any other parameters, which are necessary to reproduce the presented results. I recommend the authors specify these parameters because this information can be important to other scholars.
- Is it possible to include some penalties in the original A* algorithm (by modify its cost function or adjusting the problem formulation) to increase the distance to obstacles during the path planning? If so, I recommend the authors compare their techniques with this approach.
- Most figures have low quality and become blurry when zooming. I recommend the authors improve the quality of their figures; in particular, Figs. 5, 7, 10, 11, 12, and 15.
- The reference list has a non-uniform style and looks chaotic, and some references look incorrect (for example, [3] or [5]). I recommend the authors carefully revise the reference list according to the journal guidelines and ensure the information in each reference is complete and correct.
Minor comments:
- There are several unexplained abbreviations, for example, AGV (p. 1, l. 8), RRT (p. 1, l. 39), RRV (p. 1, l. 40), DWA (p. 2, l. 57). I recommend the authors explain each abbreviation after its first appearance, even if the abbreviation is familiar to scientific community.
- P. 3, l. 95. Notations q1 and q2 do not appear in Fig. 1.
- P. 3, Fig. 1. The figure caption does not actually explain the figure and looks inappropriate. I recommend the authors revise the caption.
- P. 3, Eq. (1) and l. 109:
- I believe there should be x instead of P or P instead of x to make authors’ notations appropriate.
- Shouldn’t there be d(P, Sj) instead of (P, Sj)?
- P. 4, l. 128. Is notation “New_Costmap” appropriate here?
- P. 4, Fig. 2. Shouldn’t there be “Customized Voronoi layers” instead of “Customize Voronoi layers”?
- P. 5, l. 163. What is n here? This parameter also repeats the one used in Sections 2.1 and 2.4, which looks inaccurate. I recommend the authors use different notations to avoid any ambiguity.
- P. 6, Fig. 4. The axes of each subfigure have no labels and units.
- P. 6, Eq. (3). Don’t the authors miss the equality signs in this equation?
- P. 6, Eq. (4). There is typo “ohter” in this equation.
- P. 6, l. 222. Shouldn’t there be lowercase notation ui instead of uppercase Ui? I recommend the authors check the rest of the paper as well.
- P. 7, l. 229. It is unclear what the authors mean under “different index iii.”
- P. 7, l. 239. Is phrase “Starting from the bottom” correct here?
- P. 7, Fig. 6. Why are the arrows two-directional?
- P. 9, Fig. 8. Is it correct to name the yellow line the set point? I recommend the authors check both the caption and the legend of the figure.
- P. 9, Eq. (7). Parameters m and t are not explained.
- P. 9, Eq. (8). Parameters t0 and t1 are not explained.
- P. 9, l. 305. Sentence “Simplify the smooth path…” starts with the verb and looks incomplete.
- P. 11, Fig. 10. It is unclear what the authors mean under the Pluinlib library.
- P. 13, Fig. 12b; p. 17, Fig. 18. What is the meaning of the red lines? I recommend the authors explain them in the text.
- P. 13, l. 397. Shouldn’t there be the reference to Fig. 13b instead if Fig. 11b?
- P. 14, l. 424. Are the AGV dimensions correct? Shouldn’t there be cm instead of mm?
- P. 14, l. 425. Phrase “odom and IMU odometer” looks incorrect.
- P. 14, l. 426. It is unclear what the authors mean under the “other equipment.” I recommend the authors be as concise as possible or omit unnecessary information.
- P. 14, l. 427. What do the authors mean under the 120 m x 50 m dimensions of the corridor?
- P. 14, l. 427–429. Sentence “As shown in Figure 14…” looks incomplete.
- P. 15, Fig. 15. The figure legend looks inaccurate because it ignores the path between points 2 and 3.
- P. 18, l. 535. The “Author Contributions” section looks incomplete.
Reviewer 2 Report
Comments and Suggestions for Authors - It would be interesting to see how the performance of the proposed path planning algorithm varies with the change in road narrowness. - More sophisticated experimental scenarios particularly the ones containing dynamic situations need to be considered to establish the over-performance of the proposed algorithm. - Also, include a critical discussion on the computational complexity of the proposed algorithm.
- Please double-check all the references and correct them wherever required. Just an example; Line 38 on Page 1 mentions "In the study by Yujun Wang et al. [8], improvements to the RRT algorithm were...". However, the first author of [8] is different as mentioned in the Bibliography section as Qisen C. - The literature review mentions some interesting path planning algorithms such s A*, RRT etc. The review can be further strengthened with other novel path planning algorithms such as IBA (Intelligent bug algorithm). - Define all the abbreviations only at their first occurrence. Later on, simply use the abbreviation. e.g. AGV in the first line of the Abstract needs to be completely defined. - Readability of the Figure 10 needs to be improved. Please remove the black background in the boxes. - Use of words "Simulation experiments" makes the discussion confusing. Please keep the simulation and experiments (on a real hardware prototype) separate. - The specifications of the AGV used for testing could better be mentioned in the form of a Specs. Table (Line 422 - Line 429). - Also, include a labeled picture of the AGV so that it conveys more useful information. - In the literature review, it would be logical to categorise the path planning algorithms in terms of their ability to handle static and dynamic situations. The discussion on dynamic algorithms could benefit from recently reported works such as A featureless approach for object detection and tracking in dynamic environments. - Please thoroughly proofread the paper for typos and other linguistic improvements
Round 2
Reviewer 1 Report
Comments and Suggestions for Authors
I would like to thank the authors for their detailed answers to my questions: the authors have addressed all the comments and improved the paper quality significantly. The paper looks almost fine, and I have just a few minor comments listed below:
- P. 1, l. 15–19. It is sufficient to write “AGV (Automated Guided Vehicle)” without the subsequent detailed explanations, which look redundant for the abstract and disrupt the sentence.
- P. 2, l. 54 and 61. The authors introduce abbreviations RFID and DON but do not spell them out explicitly.
- P. 2, l. 75–78. Similar to comment 1, it is sufficient just to spell out the DWA abbreviation without the detailed explanations.
- P. 3, Eq. (1). Shouldn’t there be R instead of Rthreshold in the left of the equation?
- P. 4, Fig. 1. The figure indicates Wroad > 2.0, Wroad = 2.0, and Wroad < 2.0, but these notations look inaccurate because there should be R > 2.0, R = 2.0, and R < 2.0.
- P. 10, l. 318. Shouldn’t the end point have coordinates (18, 18) instead of (18, 3)?
- P. 30, Table 3. It is unclear what the authors mean under the “better” path smoothness in each row of the third column of the table.
Reviewer 2 Report
Comments and Suggestions for Authors
Thanks to the authors for revising the manuscript. However, not all of my earlier comments on the initial version have not been adequately addressed. I understand that the authors are unable to consider more sophisticated experimental scenarios particularly the ones containing dynamic situations, however, please explicitly mention this as a limitation of the work or alternately, add this at the end of conclusion to list down potential enhancements. Comment 5 was related to IBA (Intelligent bug algorithm). See 'IBA: Intelligent bug algorithm – a novel strategy to navigate mobile robots autonomously'. The pivotal role of AGVs in various applciation domains including agriculture sector could benefit from the work 'Towards autonomy in agriculture: Design and prototyping of a robotic vehicle with seed selector'. In Table 3 listing the performance comparison of path planning algorithms under different road widths, can we quantify 'Better'? How much 'better' is better? Update the literature review considering notable works such as A novel goal-oriented strategy for convergence in mobile robot navigation without sub-goals constraint. In Figure showing the ath planning effect before and after smoothing, please use distinguishing markers for both quantities, say use a solid line for quantity 1, a dotted line for quantity 2. the language has been significantly improved. However, the newly added text still needs proofreading.
